# Prostaglandin $D_2$ receptor 2 downstream signaling and modulation of type 2 innate lymphoid cells from patients with asthma

**Christina Gress[1,2], Maximilian Fuchs[1], Saskia Carstensen-Aurèche[1,2], Meike Müller[1,2]⊛\*, Jens M. Hohlfeld[1,2,3]⊛\***

**1** Fraunhofer Institute of Toxicology and Experimental Medicine, Hannover, Germany, **2** German Center for Lung Research (DZL-BREATH), Hannover, Germany, **3** Department of Respiratory Medicine and Infectious Disease, Hannover Medical School, Hannover, Germany

⊛ These authors contributed equally to this work.
\* jens.hohlfeld@item.fraunhofer.de (JMH); meike.mueller@item.fraunhofer.de (MM)

**Data Availability Statement:** Raw data files (fastq. gz format) of sequenced samples and corresponding count data files obtained by RNA-Seq data analysis were published in the

## Abstract

Increased production of Prostaglandin $D_2$ ($PGD_2$) is linked to development and progression of asthma and allergy. $PGD_2$ is rapidly degraded to its metabolites, which initiate type 2 innate lymphoid cells (ILC2) migration and IL-5/IL-13 cytokine secretion in a $PGD_2$ receptor 2 ($DP_2$)-dependent manner. Blockade of $DP_2$ has shown therapeutic benefit in subsets of asthma patients. Cellular mechanisms of ILC2 activity in response to $PGD_2$ and its metabolites are still unclear. We hypothesized that ILC2 respond non-uniformly to $PGD_2$ metabolites. ILC2s were isolated from peripheral blood of patients with atopic asthma. ILC2s were stimulated with $PGD_2$ and four $PGD_2$ metabolites ($\Delta^{12}$-$PGJ_2$, $\Delta^{12}$-$PGD_2$, 15-deoxy$\Delta^{12,14}$-$PGD_2$, $9\alpha,11\beta$-$PGF_2$) with or without the selective $DP_2$ antagonist fevipiprant. Total RNA was sequenced, and differentially expressed genes (DEG) were identified by DeSeq2. Differential gene expression analysis revealed an upregulation of pro-inflammatory DEGs in ILC2s stimulated with $PGD_2$ (14 DEGs), $\Delta^{12}$-$PGD_2$ (27 DEGs), 15-deoxy$\Delta^{12,14}$-$PGD_2$ (56 DEGs) and $\Delta^{12}$-$PGJ_2$ (136 DEGs), but not with $9\alpha,11\beta$-$PGF_2$. Common upregulated DEGs were i.e. ARG2, SLC43A2, LAYN, IGFLR1, or EPHX2. Inhibition of $DP_2$ via fevipiprant mainly resulted in downregulation of pro-inflammatory genes such as DUSP4, SPRED2, DUSP6, ETV1, ASB2, CD38, ADGRG1, DDIT4, TRPM2, or CD69. DEGs were related to migration and various immune response-relevant pathways such as "*chemokine (C-C motif) ligand 4 production*", "*cell migration*", "*interleukin-13 production*", "*regulation of receptor signaling pathway via JAK-STAT*", or "*lymphocyte apoptotic process*", underlining the pro-inflammatory effects of $PGD_2$ metabolite-induced immune responses in ILC2s as well as the anti-inflammatory effects of $DP_2$ inhibition via fevipiprant. Furthermore, $PGD_2$ and metabolites showed distinct profiles in ILC2 activation. Overall, these results expand our understanding of $DP_2$ initiated ILC2 activity.

ArrayExpress Archive of Functional Genomics Data (https://www.ebi.ac.uk/biostudies/arrayexpress/studies/E-MTAB-13623).

**Funding:** This research was supported by the German Center for Lung Research. The funders had no role in study design, data collection and analysis, decision to publish, or preparation of the manuscript.

**Competing interests:** The authors have declared that no competing interests exist.

**Abbreviations:** Adj, Adjusted; BAL, Bronchoalveolar lavage; BMI, Body mass index; CCL, C-C Motif Chemokine Ligand; CD, Cluster of differentiation; CRTH2, Chemoattractant receptor-homologous molecule expressed on Th2 cells; DEG, Differential expressed genes; DMSO, Dimethylsulfoxide; $DP_2$, $PGD_2$ receptor 2; $EC_{70}$, 70% of maximal effective concentration; IL, Interleukin; ILC2, Type 2 innate lymphoid cells; Log2FC, Log 2 fold change; M, Molar; PBMCs, Peripheral blood mononuclear cells; PCA, Principal component analysis; $PGD_2$, Prostaglandin $D_2$.

## Introduction

Approximately 262 million people worldwide were affected by asthma in 2019, leading to about 450,000 deaths per year with increasing tendency [1]. Asthma causes various clinical symptoms such as shortness of breath, chest tightness, wheezing and coughing [1]. There is still an enormous need of new drugs to treat particularly severe, uncontrolled asthmatic patients not responding sufficiently to current available treatment options [2,3]. The discovery of type 2 innate lymphoid cells (ILC2), which were found to be increased in the airways of severe asthmatics [4,5], led to a reassessment of the pathogenesis of allergic asthma. In addition, development and progression of asthma and allergy is linked to increased production of the ILC2 activating hormone prostaglandin $D_2$ ($PGD_2$) [6–9]. $PGD_2$ is mainly secreted by epithelial cells as well as mast cells and ILC2s, which thereby binds to the $PGD_2$ receptors 1 ($DP_1$) and 2 ($DP_2$, synonym: CRTH2) presumably expressed by Th2 cells, eosinophils and ILC2s [10–12]. While $DP_1$ signaling is predominantly associated with anti-inflammatory effects such as neuronal protection, inhibition of immune cell function and reduced cell migration [13], activation of $DP_2$ results in pro-inflammatory outcomes such as increased migration and type 2 cytokine secretion [11,14,15]. However, some mechanisms require cooperative signaling between $DP_1$ and $DP_2$, i.e., $PGD_2$ induced leukotriene $C_4$ synthesis in eosinophils [16]. Furthermore, thromboxane (TP) receptors [17,18] and peroxisome proliferator activated receptor gamma (PPAR-γ) [19] are also activated by $PGD_2$. Activation of TP results in smooth muscle contraction [18,20], while signaling via PPAR-γ promotes type 2 allergic responses in mice [21,22] and is involved in modulation of immune and inflammatory responses [23–25]. PPAR-γ receptors are expressed on ILC2s and are further increased following interleukin-33 (IL-33) stimulation [22,26], but there is no clear evidence to date showing the expression of TP receptors on ILC2s.

Within 120 minutes, more than 92% of $PGD_2$ is transformed either enzymatically or spontaneously to several $PGD_2$ metabolites such as $\Delta^{12}$-$PGD_2$, $\Delta^{12}$-$PGJ_2$, 15-deoxy-$\Delta^{12,14}$-$PGD_2$, 15-deoxy-$\Delta^{12,14}$-$PGJ_2$ and 9α,11β-$PGF_2$ [27,28]. While most of the $PGD_2$ metabolites have mainly pro-inflammatory effects [15,29], for 15-deoxy-$\Delta^{12,14}$-$PGJ_2$ also anti-inflammatory properties were reported [30,31]. These $PGD_2$ metabolites have selective binding affinity to $DP_2$ over $DP_1$ and thus may play a role in inflammatory immune responses via $DP_2$ signaling [13,32]. We have previously investigated the effects of $PGD_2$ and its metabolites 13,14-dihydro-15-keto-$PGD_2$, $\Delta^{12}$-$PGD_2$, 15-deoxy-$\Delta^{12,14}$-$PGD_2$, 9α,11β-$PGF_2$, $PGJ_2$, $\Delta^{12}$-$PGJ_2$ and 15-deoxy-$\Delta^{12,14}$-$PGJ_2$ on ILC2 activity in presence or absence of the selective $DP_2$ antagonist fevipiprant demonstrating that all selected $PGD_2$ metabolites except 9α,11β-$PGF_2$ induced ILC2 migration in a $DP_2$-dependent manner with $EC_{50}$ values ranging from 17.4 to 91.7 nM [15]. Compared to $PGD_2$, ILC2 migration was enhanced in the presence of 15-deoxy-$\Delta^{12,14}$-$PGD_2$, $\Delta^{12}$-$PGJ_2$ and 15-deoxy-$\Delta^{12,14}$-$PGJ_2$. Additionally, we found that $PGD_2$ metabolites induce cytokine secretion of IL-5 and IL-13 by ILC2s in a $DP_2$-dependent manner, whereby 9α,11β-$PGF_2$ showed reduced potency compared to the other metabolites (IL5 range: 108.1 to 526.9 nM, IL-13 range: 125.2 to 788.3 nM). ILC2s stimulated with $\Delta^{12}$-$PGD_2$ or 15-deoxy-$\Delta^{12,14}$-$PGD_2$ secreted numerical elevated levels of the type 2 cytokines IL-5 and IL-13 compared to $PGD_2$ stimulated ILC2s [15]. As the $PGD_2$ metabolites showed different potencies to activate ILC2s, these may lead to diverse biological consequences.

Furthermore, inhibition of the $DP_2$ signaling pathway may be a potential drug target for severe, uncontrolled asthmatic patients [3,33]. Fevipiprant is a potent and selective $DP_2$ antagonist, which as such is well-tolerated and has shown anti-inflammatory effects such as reduced eosinophilic airway inflammation and declined airway smooth muscle mass in phase 2 clinical trials with patients suffering from severe, uncontrolled asthma [34–36]. In none of the phase 3

studies conducted so far, fevipiprant treatment of patients with severe asthma resulted in statistically significant improved outcomes [37–39]. However, in two phase 3 clinical trials modest reductions of asthma exacerbation rates in the high fevipiprant dose group were observed [38]. *In vitro* experiments showed that fevipiprant inhibited cell aggregation, migration and cytokine secretion by $PGD_2$ or $PGD_2$ metabolites activated human ILC2s [15,40]. However, underlying cellular mechanisms of ILC2 activity via the $DP_2$ receptor in response to $PGD_2$ and $PGD_2$ metabolites are still unclear. We hypothesized that ILC2 respond non-uniformly to $PGD_2$ metabolites.

Therefore, transcriptomics of ILC2s isolated from blood of asthma patients and incubated with $PGD_2$ and four $PGD_2$ metabolites ($\Delta^{12}$-$PGJ_2$, $\Delta^{12}$-$PGD_2$, 15-deoxy$\Delta^{12,14}$-$PGD_2$, 9α,11β-$PGF_2$) in presence or absence of the selective $DP_2$ antagonist fevipiprant were analyzed to further our understanding of $DP_2$ initiated ILC2 activity in asthma. These experiments follow and add to previously published *in vitro* experiments using the same patient material, which have shown that $PGD_2$ metabolites induce ILC2 migration and IL-5/IL-13 cytokine secretion in a $DP_2$-dependent manner [15].

## Materials and methods

### Study design

Four volunteers with mild intermittent atopic asthma (three male/ one female; average age: 28.3 ± 4.0 years; BMI: 19–32 $kg/m^2$, with a clinically manifest allergy against house dust mite, no corticosteroids for > four weeks) were enrolled into the study between 9th November 2020 and 18th February 2021. Whole blood (500 ml) was collected in 3.8% trisodium citrate and was processed within one hour after blood withdrawal. ILC2s were isolated from the peripheral blood, cultured and expanded. On the one hand, ILC2s were used for migration experiments towards $PGD_2$ and its metabolites with or without the selective $DP_2$ antagonist fevipiprant [15], and on the other hand, ILC2s were stimulated with $PGD_2$ and its metabolites with or without fevipiprant within the culture plates. Total RNA from the stimulated cells within the culture plates was sequenced for transcriptomic analysis (focus of this manuscript), and IL-5/IL-13 cytokine concentrations were measured in the supernatant [15].

### Ethics approval and consent to participate

The protocol was approved by the Ethics Committee of Hannover Medical School, Hannover, Germany under reference 839–2010. The study was conducted at the Fraunhofer Institute for Toxicology and Experimental Medicine, Hannover, Germany in accordance with the Declaration of Helsinki and the International Council for Harmonisation (ICH) Harmonised Tripartite Guideline for Good Clinical Practice. Written informed consent was obtained from all participants after they were fully informed about all study aspects before any study-related procedures.

### Reagents

$PGD_2$, $\Delta^{12}$-$PGJ_2$, $\Delta^{12}$-$PGD_2$, 15-deoxy-$\Delta^{12,14}$-$PGJ_2$, 9α,11β-$PGF_2$ were purchased from Cayman Chemicals (Biomol GmBH, Hamburg, Germany). Fevipiprant (GST0000013789) was provided by Novartis Pharma AG (Basel, Switzerland). Reagents were dissolved in sterile-filtered Hybri-Max dimethylsulfoxide (DMSO, Sigma-Aldrich, Taufkirchen, Germany).

### Type 2 innate lymphoid cell isolation and cell culture

Details of ILC2 isolation and cell culture as well as reagent information were previously described in detail elsewhere [15]. Briefly, peripheral blood mononuclear cells (PBMCs) were

isolated from 500 mL peripheral blood of patients with atopic asthma (n = 4) using Ficoll density gradient centrifugation. ILC2 were enriched by depletion of T cells, B cells and monocytes using CD3-, CD14- and CD19-MACS separation beads (Miltenyi Biotech, Bergisch Gladbach, Germany), respectively. Cells were stained for 15 min at RT with a PerCP-Cy5.5-labeled lineage cocktail 1 (CD4, CD8, CD14, CD16, CD19, CD34, CD123, FcεRI), a FITC-labeled lineage cocktail 2 (CD11b, CD56), CD3-BV510, CD127-BV421, CD45-Alexa Fluor 700 and CD294-PE. For more information on flow cytometric antibodies view supplements of our previous publication [15]. Afterwards, ILC2 cells (CD45$^+$, CD4$^-$, CD8$^-$, CD14$^-$, CD16$^-$, CD19$^-$, CD34$^-$, CD123$^-$, FcεRI$^-$, CD11b$^-$, CD56$^-$, CD3$^-$, CD127$^+$, CD294$^+$) were sorted with an FACS ARIA Fusion (BD Bioscience) into 96 U bottom well plates (100 cells/ well). The corresponding ILC2 gating strategy is shown in S1 Fig. Sorted cells were expanded with human feeder PBMCs (100,000/ well; 37˚C, 5% $CO_2$) in culture medium (RPMI 1640 Glutamax medium, 1% Pen/Strep, 10% *h.i.* human AB serum, 25 mM HEPES, 100 U/ ml rh-IL-2, 25 ng/ ml rh-IL-4, and 5 μg/ ml phytohemagglutinin-M (PHA-M)) for three to five weeks.

## Type 2 innate lymphoid metabolite incubation and sample processing

Cells (3–15 x10$^4$/ well) were incubated with 1 μM fevipiprant for 1 h and afterwards with the EC$_{70}$ concentration of PGD$_2$ (EC$_{70}$ = 348.0 nM) or the selected PGD$_2$ metabolites Δ$^{12}$-PGD$_2$ (EC$_{70}$ = 364.0 nM), Δ$^{12}$-PGJ$_2$ (EC$_{70}$ = 759.0 nM), 15-deoxy-Δ$^{12,14}$-PGD$_2$ (EC$_{70}$ = 446.0 nM) or 9α,11β-PGF$_2$ (EC$_{70}$ = 520.6 nM) for 24 h (U-bottom 96-well plates, 37˚C, 5% $CO_2$) in culture medium without IL-2, IL-4 and PHA-M. EC$_{70}$ concentrations of PGD$_2$ and metabolites were calculated based on preliminary titration experiments, where IL-5/IL-13 cytokine secretion of ILC2s in the presence of ascending PGD$_2$ and metabolite concentrations was investigated (results are shown in Fig 4 of our previous publication [15]). For fevipiprant a concentration of 1 μM was chosen, because metabolite-induced cytokine secretion was completely abolished at this concentration (results are shown in Fig 5 of our previous publication [15]). Afterwards, cell samples were centrifuged (5 min, 300 x g, RT), stabilized in RNAprotect Cell Reagent (Qiagen, Venlo, The Netherlands) and stored at -80˚C until further handling. Total RNA was isolated from ILC2 using the Rneasy Plus Mini Kit (Qiagen, Venlo, The Netherlands) following the manufacturer's protocol. Concentration and purity were determined in the isolated RNA solution via absorbance measurement using a NanoDrop device. Eluted RNA was stored at -80˚C. For RNA-Sequencing (RNA-Seq) library preparation was performed using the NEB-Next Ultra™ II Directional RNA Library Prep Kit (New England Biolabs, Frankfurt am Main, Germany) following the manufacturer's instructions, and all samples were sequenced paired-end and strand-specific on the NovaSeq 6000 (Illumina, San Diego, USA) with a sequencing-depth of 100 million reads per sample.

## Processing of RNA-Seq raw data

Sequenced data were analyzed using the Galaxy web platform (usegalaxy.eu) [41]. Default settings were used for the tool applications, unless otherwise mentioned. For quality control *FastQC Galaxy Version 0.72* was applied to the raw data and reports were checked for "per base sequence quality", "overrepresented sequences" and "adapter content" [42]. Data with poor quality in "per base sequence quality" or "adapter content" were excluded from further analysis. Data with "overrepresented sequences" were trimmed via *fastp Galaxy Version 0.20.*1 [43] and checked again for quality using *FastQC*. Reads were mapped to the human GRCh38 reference genome (**https://www.gencodegenes.org/human/releases.html**) using the Gencode main annotation file via *RNA Star Galaxy Version 2.7.8a* [44]. From the output with mapped sequences, the number of reads per annotated genes were determined using *FeatureCounts*

*Galaxy Version 2.0.1* [45]. Differential expressed gene (DEG) analysis was performed using R (Version R 4.3.1). To remove unwanted variation the control gene method *RUVSeq Version 1.34.0* was applied to raw read counts [46]. Using *DESeq2 Version 1.40.2* counts were normalized, PCA-Plots were created, and differential expression was calculated using paired sample analysis [47]. Adjusted (adj.) p-values were calculated for multiple testing by *DESeq2* using the Benjamini-Hochberg procedure which controls false discovery rate (FDR). Gene names were determined in the manuscript based on ENSEMBL release 103 (February 2021) [48]. DEGs were defined by the following criteria: adjusted p-value $\leq$ 0.05, BaseMean $\geq$ 5, and Log2 fold change ($|Log2FC|$) $\geq$ 0.58 for deregulated genes.

In addition, genes associated with the PPAR-$\gamma$ signaling pathway according to the KEGG database (**https://www.genome.jp/pathway/hsa03320**) were examined for DEGs that were induced by ILC2 stimulation with $PGD_2$ or its metabolites compared to unstimulated ILC2s (DEGs were defined as described above; results are shown in S1 Table).

### Visualization of processed RNA-Seq data

Data were visualized using GraphPad Prism 9.0.1. Gene set enrichment analysis was performed with differentially expressed genes using *g:Profiler* (Version 2019) to determine enriched pathways (KEGG_pathways), and biological processes (GOTERM_BP_DIRECT) [49]. Adj. p-values were calculated for multiple testing by *g:Profiler* using the Benjamini-Hochberg procedure which controls FDR. Significantly enriched pathways showed an adj. p-value $\leq$ 0.05. Selected immune response-relevant enriched pathways were visualized using the R package *ggplot2 Version 3.4.4* (entire dataset is given in the supplements).

## Results

### Type 2 immune response- as well as migration-related genes were upregulated in ILC2s stimulated with $PGD_2$, $\Delta^{12}PGD_2$, $\Delta^{12}PGJ_2$ and 15-deoxy$\Delta^{12,14}$-$PGD_2$, but not in 9$\alpha$,11$\beta$-$PGF_2$ stimulated ILC2s

Differential gene expression analysis resulted in 14 differentially expressed genes (DEGs) for $PGD_2$-, 27 DEGs for $\Delta^{12}$-$PGD_2$-, 136 DEGs for $\Delta^{12}$-$PGJ_2$- and 56 DEGs for 15-deoxy-$\Delta^{12,14}$-$PGD_2$-, and none DEGs for 9$\alpha$,11$\beta$-$PGF_2$-stimulated cells, when compared to unstimulated cells (Fig 1). Common upregulated genes were ARG2, SLC43A2, LAYN, IGFLR1 and EPHX2. Inhibition of the $DP_2$ receptor via fevipiprant and following stimulation with $PGD_2$ or metabolites resulted in 97 DEGs for $PGD_2$-, 109 DEGs for $\Delta^{12}$-$PGD_2$-, 116 DEGs for $\Delta^{12}$-$PGJ_2$-, 46 DEGs for 15-deoxy-$\Delta^{12,14}$-$PGD_2$-, and 256 DEGs for 9$\alpha$,11$\beta$-$PGF_2$-stimulaed ILC2s compared to stimulated ILC2 without fevipiprant (Fig 2). Genes were mainly downregulated including common genes such as DUSP4, SPRED2, DUSP6, ETV1, ASB2, CD38, ADGRG1, DDIT4, TRPM2 and CD69. Detailed results for each gene of all samples are provided in S2 and S3 Tables, including calculated log2 Fold change (log2FC) with corresponding adjusted p-values. As $PGD_2$ and its metabolites 15-deoxy-$\Delta^{12,14}$-$PGJ_2$ and $\Delta^{12}$-$PGJ_2$ from the J-series can bind and activate the PPAR-$\gamma$ receptor [19,21,50], we have additionally investigated the expression of genes that are related to the PPAR-$\gamma$ signaling pathway. However, none of the PPAR-$\gamma$ related genes were significantly differentially expressed in $PGD_2$ or metabolite stimulated ILC2s compared with unstimulated ILC2s (S1 Table).

### $PGD_2$ and its metabolites show distinct profiles in ILC2 activation

Gene set enrichment analysis with identified DEGs ($|log2FC| \geq$ 0.58, adj. p-value $\leq$ 0.05) revealed multiple pathways that play a role in immune responses (main results in Fig 3, all

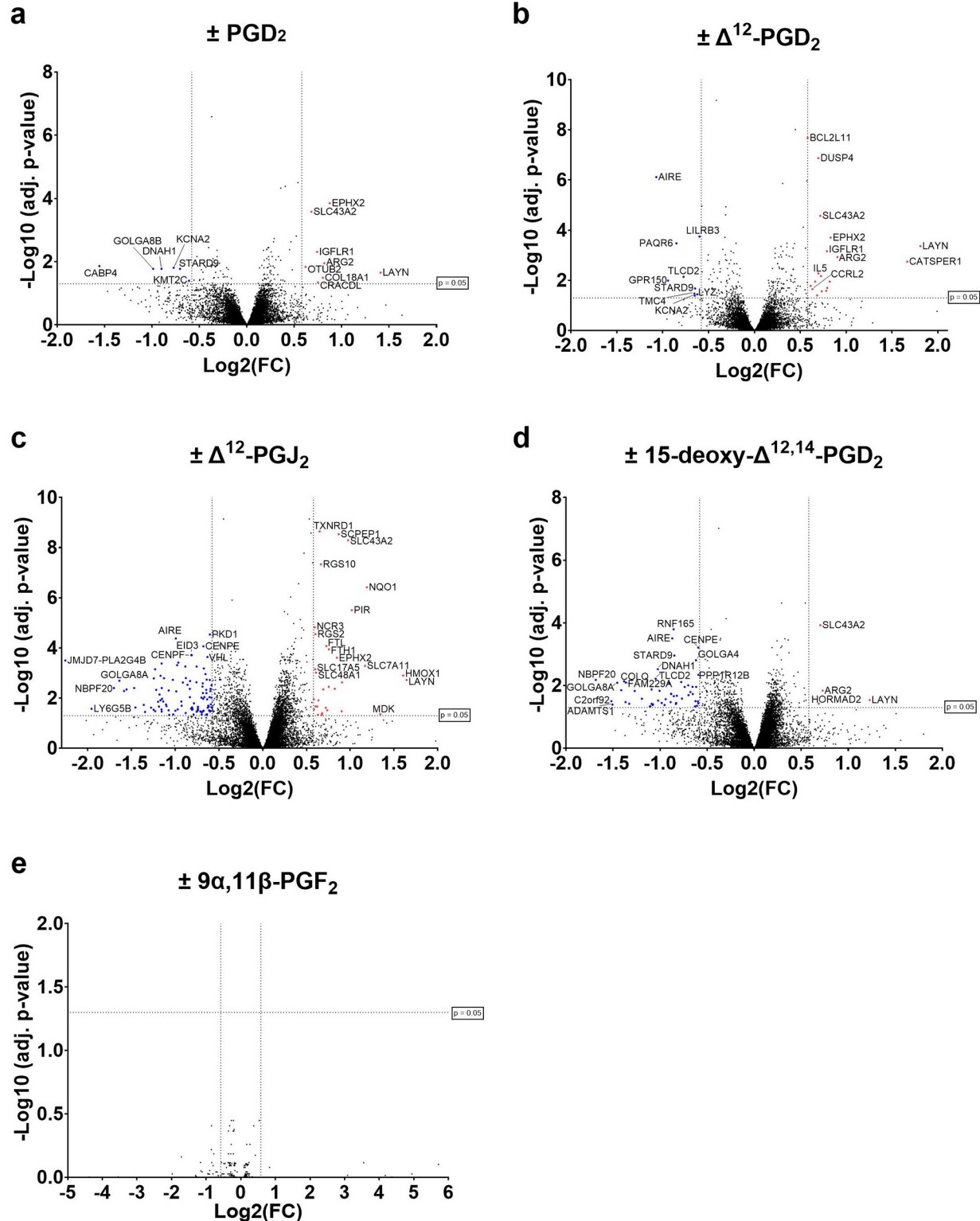

**Fig 1.** DEG analysis of ILC2s stimulated with (a) $PGD_2$, (b) $\Delta^{12}$-$PGD_2$, (c) $\Delta^{12}$-$PGJ_2$, (d) 15-deoxy-$\Delta^{12,14}$-$PGD_2$ and (e) 9α,11β-$PGF_2$ versus unstimulated ILC2s. DEGs are highlighted in red for upregulated genes (adj. p-value ≤ 0.05, log2FC ≥ 0.58, BaseMean ≥ 5) and in blue for downregulated genes (adj. p-value ≤ 0.05, log2FC ≤ -0.58, BaseMean ≥ 5).

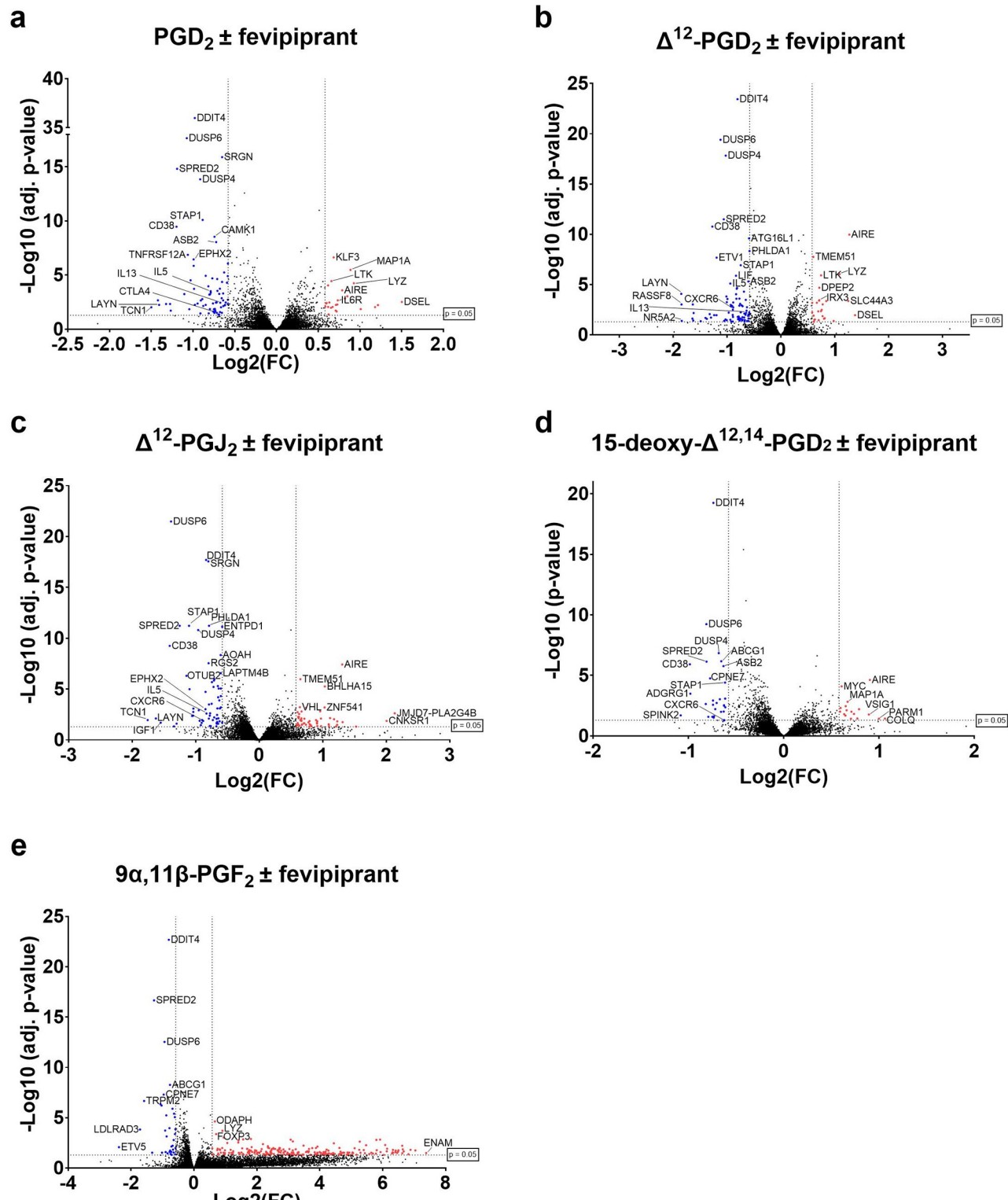

**Fig 2.** DEG analysis of ILC2s incubated with fevipiprant versus ILC2s stimulated with (a) $PGD_2$, (b) $\Delta^{12}$-$PGD_2$, (c) $\Delta^{12}$-$PGJ_2$, (d) 15-deoxy-$\Delta^{12,14}$-$PGD_2$ and (e) 9α,11β-$PGF_2$. DEGs are highlighted in red for upregulated genes (adj. p-value $\leq$ 0.05, log2FC $\geq$ 0.58, BaseMean $\geq$ 5) and in blue for downregulated genes (adj. p-value $\leq$ 0.05, log2FC $\leq$ -0.58, BaseMean $\geq$ 5).

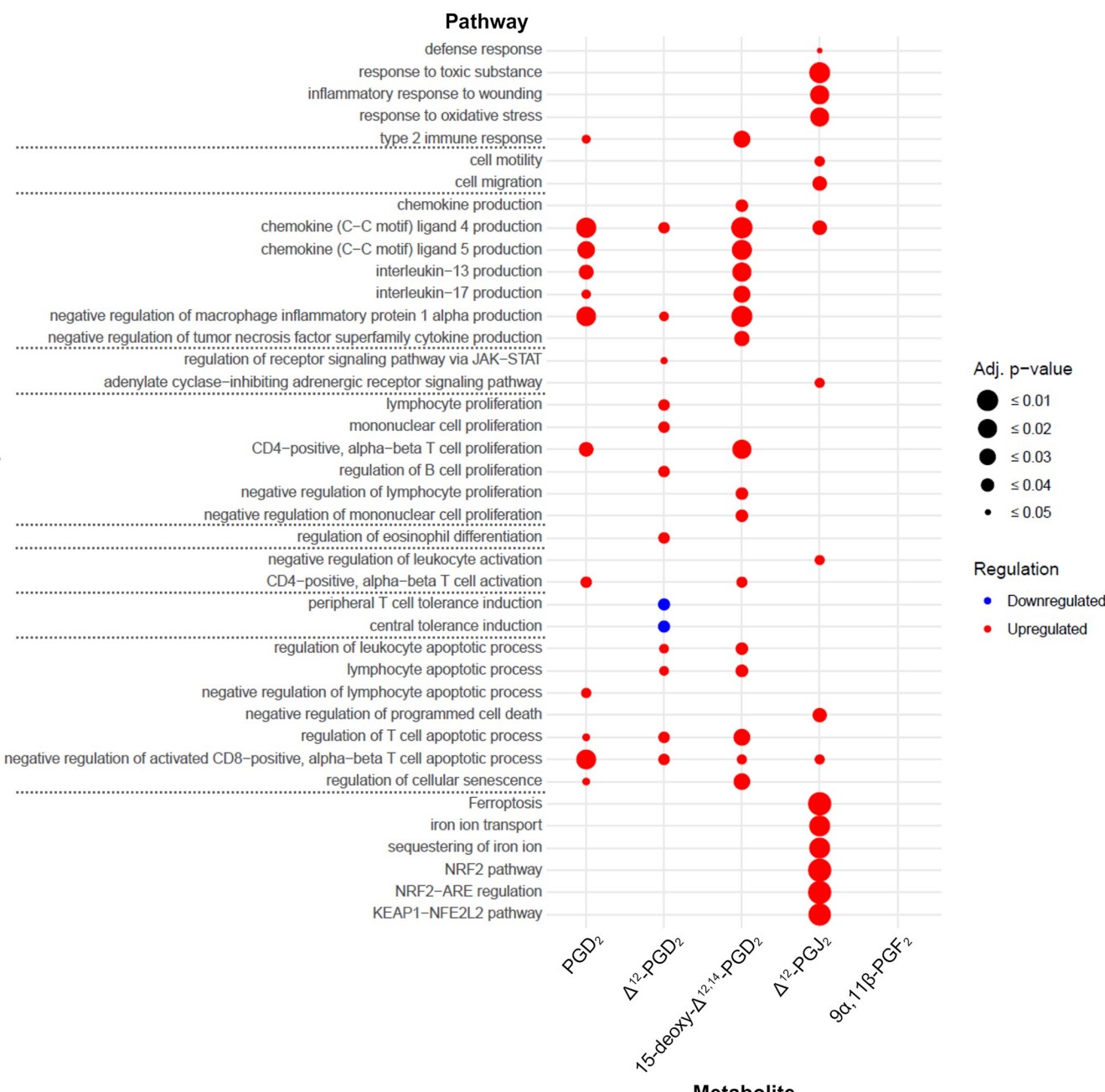

**Fig 3. Gene set enrichment analysis with identified DEG ($|\log2FC| \geq 0.58$, adj. p-value $\leq 0.05$) of metabolite activated ILC2s compared to unstimulated ILC2s (red = upregulated pathways, blue = downregulated pathways, increasing circle size corresponds to increasing significance).**

results in S4 and S5 Tables). Unexpectedly, "*cell migration*" and "*cell motility*" pathways were only enhanced in $\Delta^{12}$-PGJ$_2$ stimulated ILC2. "*Chemokine production*" including C-C Motif Chemokine Ligand 4 (CCL4), CCL5, IL-13 as well as IL-17 production was elevated in ILC2s stimulated with in particular PGD$_2$ and 15-deoxy-$\Delta^{12,14}$-PGD$_2$. ILC2s activated with $\Delta^{12}$-PGD$_2$ showed an upregulation of genes associated with the "*regulation of receptor signaling pathway via JAK-STAT*" and "*regulation of eosinophil differentiation*". In addition, identified

DEGs of $PGD_2$, $\Delta^{12}$-$PGD_2$, and 15-deoxy-$\Delta^{12,14}$-$PGD_2$, but not $\Delta^{12}$-$PGJ_2$ or $9\alpha,11\beta$-$PGF_2$ stimulated cells contributed to regulation of lymphocyte and mononuclear cell proliferation. Furthermore, gene set enrichment analysis revealed upregulation of apoptosis-relevant pathways in ILCs stimulated with $PGD_2$, $\Delta^{12}$-$PGD_2$, $\Delta^{12}$-$PGJ_2$ and 15-deoxy-$\Delta^{12,14}$-$PGD_2$, but not with $9\alpha,11\beta$-$PGF_2$. Only $\Delta^{12}$-$PGJ_2$ stimulated ILC2 showed an upregulation of the "*adenylate cyclase-inhibiting adrenergic receptor signaling pathway*" as well as ferroptosis related pathways. None DEGs were identified in $9\alpha,11\beta$-$PGF_2$ stimulated cells, and therefore no $9\alpha,11\beta$-$PGF_2$ regulated pathways could be found (Fig 3). As expected based on *in vitro* experiments [15], pre-incubation of ILC2 with the selective $DP_2$ antagonist fevipiprant followed by stimulation with $PGD_2$, $\Delta^{12}$-$PGD_2$, $\Delta^{12}$-$PGJ_2$ or 15-deoxy-$\Delta^{12,14}$-$PGD_2$ attenuated the "*immune response*" (Fig 4). This included various pathways regarding "*cell migration*", "*cytokine production*" (e.g., IL-13, CCL4, CCL5), signaling cascades (e.g., "*MAPK cascade*", "*ERK1 and ERK2 cascade*"), as well as regulation of cell activation, proliferation, differentiation, phagocytosis, and death (Fig 4). In contrast, ILC2 incubation with fevipiprant prior to $9\alpha,11\beta$-$PGF_2$ stimulation resulted in downregulation of only some general immune response-related pathways such as "*cell communication*" while "*cell migration/ adhesion*" were upregulated in these cells.

In addition to DEG and gene set enrichment analysis, principal component analysis (PCA) was performed using all ILC2 samples (unstimulated ILC2s, ILC2 ± $PGD_2$/metabolites ± fevipiprant). Unexpectedly, the samples were not clustered by groups (unstimulated vs. metabolites vs. fevipiprant), but by participants (S2 Fig). The top three genes of the principal component 1 (PC1) were TRDV3, ITGAD and ERAP2, while PC2 was dominated by expression of the top three genes TRDV2, RPS4Y1 and SELL. However, principal component analysis of ILC2s sorted by the four individual participants resulted in group-specific clustering (S3 Fig).

## Discussion

We have previously shown that $PGD_2$ metabolites induce eosinophil shape change, ILC2 cell migration and type 2 cytokine secretion via the $DP_2$ receptor [15]. However, the corresponding underlying cellular mechanisms of $DP_2$ and potential differences between $PGD_2$- and $PGD_2$ metabolites-induced ILC2 activity are still unclear.

As expected, and in accordance with our previous findings [15], ILC2 stimulation with $PGD_2$ or $PGD_2$ metabolites (except $9\alpha,11\beta$-$PGF_2$) resulted in upregulation of pro-inflammatory genes. ILC2 stimulation resulted in different numbers of DEG depending on the used metabolite, which could indicate that $PGD_2$ and its metabolites may have different biological purposes in ILC2 activation. In contrast to the other examined metabolites, stimulation with $9\alpha,11\beta$-$PGF_2$ did not induce DEGs. Therefore, it is possible that $9\alpha,11\beta$-$PGF_2$ is a degradation product. On the other hand, the used $9\alpha,11\beta$-$PGF_2$ concentrations may not have been sufficient to compensate the 130 fold reduced $DP_2$ binding affinity of $9\alpha,11\beta$-$PGF_2$ compared to $PGD_2$ [32]. Compared with the other metabolites, stimulation with $\Delta^{12}$-$PGJ_2$ led to the highest DEG number (136 DEGs) followed by 15-deoxy-$\Delta^{12,14}$-$PGD_2$ (56 DEG), $\Delta^{12}$-$PGD_2$ (27 DEGs) and then $PGD_2$ (14 DEGs). Common upregulated genes were e.g. important for T cell immunobiology (ARG2 [51], SLC43A2 [52], IGFLR1 [53]), involved in cell migration (LAYN [54]) or known to be involved in cigarette smoke-induced pulmonary inflammation and autophagy in mice (EPHX2 [55]). Some upregulated genes of this study were previously identified as potential targets for cancer therapeutics (LAYN, IGFLR1 [56]). As LAYN and IGFLR1 appear to play a role in the immune response of asthmatic patient-derived ILC2s, these genes could also have the potential to be valuable targets for the treatment of asthma. $DP_2$ inhibition via fevipiprant led to downregulation of in particular migration-related genes (DUSP6 [57], ETV1 [58], ASB2 [59,60], CD38 [61], ADGRG1 [62], DDIT4 [63], TRPM2 [64]). Furthermore, genes

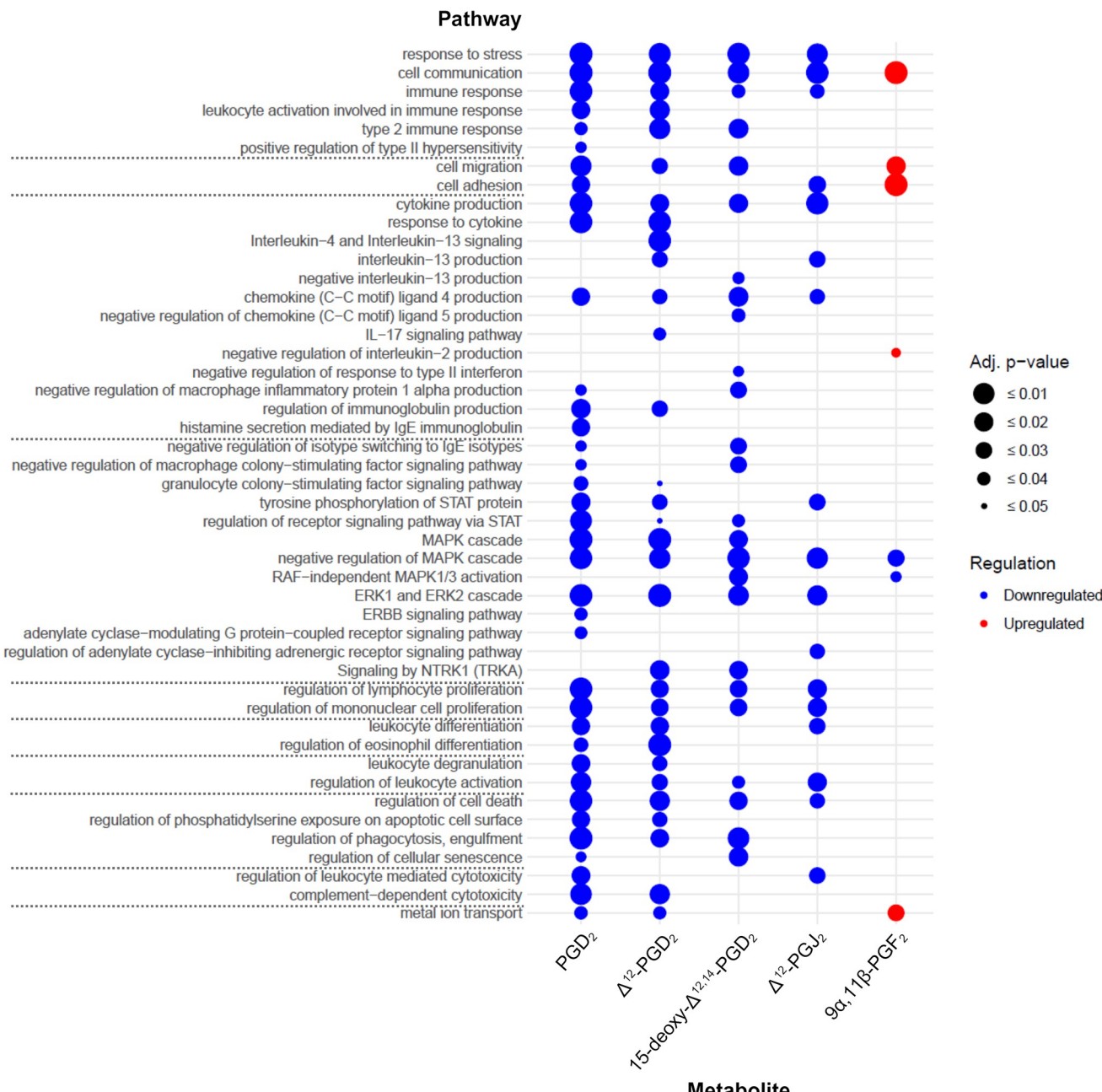

**Fig 4. Gene set enrichment analysis with identified DEGs ($|log2FC| \geq 0.58$, adj. p-value $\leq 0.05$) of metabolite activated ILC2s pre-incubated with the selective DP$_2$ antagonist fevipiprant compared to metabolite activated ILC2s without fevipiprant stimulation (red = upregulated pathways, blue = downregulated pathways, increasing circle size corresponds to increasing significance).**

that are involved in the MAPK- and ERK signaling pathway (DUSP4 [65], SPRED2 [66]) or known to play a role in pathogenesis of allergic airway inflammation (CD69 [67]) were down-regulated by DP$_2$ inhibition.

The prominent pro-inflammatory effects of PGD$_2$ metabolite stimulation as well as anti-inflammatory effects of the selective DP$_2$ antagonist fevipiprant were also mirrored by gene set enrichment analysis. With the exception of stimulation with 9α,11β-PGF$_2$, immune response-

related pathways were generally upregulated in $PGD_2$ or metabolite stimulated ILC2s, while $DP_2$ inhibition via fevipiprant led to downregulation of these pathways. However, $PGD_2$ and its metabolites showed variabilities in their potential to activate pro-inflammatory pathways. Metabolites of the D-series ($PGD_2$, $\Delta^{12}$-$PGD_2$, 15-deoxy-$\Delta^{12,14}$-$PGD_2$) have shown a comparable activation pattern in ILC2s including production of different cytokines, regulation of cell proliferation and apoptotic processes. Thereby 15-deoxy-$\Delta^{12,14}$-$PGD_2$ induced an enhanced, while $\Delta^{12}$-$PGD_2$ induced an attenuated immune response in ILC2s compared to the origin molecule $PGD_2$. In contrast, ILC2 stimulation with $\Delta^{12}$-$PGJ_2$ primarily induced "*cell motility/ migration*" and "*Ferroptosis*"-related pathways such as "*iron ion transport*", "*NRF2 pathway*" or "*KEAP-NFE2L2 pathway*". The reason why only $\Delta^{12}$-$PGJ_2$ was able to induce ferroptosis-related pathways in ILC2s, although all metabolites bind to the same receptor $DP_2$, remains speculative.

The different biological responses of ILC2s to stimulation with $PGD_2$ or its metabolites could be attributed to additional interactions with other relevant receptors. $PGD_2$ has been shown to activate $DP_1$ [13] and TP receptors [17,18]. However, as $DP_1$ is only expressed at low levels [12] and TP has not yet been clearly reported to be expressed on ILC2 cells, it is unlikely that these signaling pathways are involved in ILC2 activation. Another potentially relevant receptor is PPAR-γ, which can be activated by $PGD_2$ [19] or its metabolites $\Delta^{12}$-$PGJ_2$ and 15-deoxy-$\Delta^{12,14}$-$PGJ_2$ from the J-series [19,21,50]. To the best of our knowledge, there is no literature elucidating effects of $PGD_2$ metabolites from the D-series on the PPAR-γ receptor. Therefore, the different activation pattern of $\Delta^{12}$-$PGJ_2$ compared to the metabolites from the D- and F-series, may at least be partly attributed to signaling via PPAR-γ. However, genes that are related to the PPAR-γ signaling pathway were not differentially expressed in ILC2s stimulated with $\Delta^{12}$-$PGJ_2$ or other metabolites compared to unstimulated ILC2s (S1 Table). To fully clarify whether the PPAR-γ signaling pathway is involved in ILC2 activation in response to $PGD_2$ or its metabolites, experiments with a PPAR-γ antagonist such as GW9662 would need to be performed. However, as inhibition of $DP_2$ via fevipiprant showed powerful inhibitory effects, we assume that ILC2 activation via $DP_2$ is the most prominent signaling pathway.

Overall, the observations described in this publication are consistent with our previously described in vitro experiments from the same study, where cell migration was reported to be strongest for the J-series metabolites, cytokine secretion was strongest for the D-series metabolites and 9α,11β-$PGF_2$ only showed barely effects [15]. In addition to the previously performed experiments, transcriptomic analysis revealed new relevant pathways which need to be confirmed *in vitro* and *in vivo*.

Interestingly, the inhibitory effects of fevipiprant seemed to be more prominent in cells activated with metabolites from the D-series ($PGD_2$, $\Delta^{12}$-$PGD_2$, 15-deoxy-$\Delta^{12,14}$-$PGD_2$). Since ILC2 stimulation with 9α,11β-$PGF_2$ did not induce changes in the transcriptomic profile compared to unstimulated cells, the lack of down-regulated pathways after fevipiprant incubation was expected. On the contrary, pathways such as "*cell communication*, "*cell migration*" or "*cell adhesion*" were upregulated in 9α,11β-$PGF_2$ stimulated cells that were preincubated with fevipiprant. The reason for this remains unclear.

$PGD_2$ and metabolites bind $DP_2$ with higher affinity than $DP_1$ and in the following rank order of potency: $PGD_2$ ($K_i$ = 2.4) > $\Delta^{12}$-$PGJ_2$ ($K_i$ = 6.8) > $\Delta^{12}$-$PGD_2$ ($K_i$ = 7.63) >>> 9α,11β-$PGF_2$ ($K_i$ = 315.0) (unknown $K_i$ for 15-deoxy-$\Delta^{12,14}$-$PGD_2$) [32,68]. As outlined above, the low $DP_2$ affinity of 9α,11β-$PGF_2$ could therefore explain the reduced ability of this metabolite to activate ILC2s. However, the binding affinity of the other metabolites is not in line with the potency to induce DEGs in ILC2s, where the rank order was $\Delta^{12}$-$PGJ_2$ > 15-deoxy-$\Delta^{12,14}$-$PGD_2$ > $\Delta^{12}$-$PGD_2$ > $PGD_2$ > 9α,11β-$PGF_2$, and therefore does not explain the different ILC2 activation patterns. Another reason for the different potencies of $PGD_2$ and its metabolites to

activate ILC2s could be their local distributions in human tissue. However, the current knowledge about $PGD_2$ metabolism in human tissue is only present to a limited extend. In plasma, $PGD_2$ is presumably degraded into $\Delta^{12}$-$PGJ_2$ and $\Delta^{12}$-$PGD_2$, while only small amounts of 15-deoxy-$\Delta^{12,14}$-$PGD_2$ are formed [27]. Increased $9\alpha,11\beta$-$PGF_2$ could be measured in plasma and urine of asthmatic patients following allergen challenge [69]. $PGD_2$ was found in BAL [9] and in induced sputum [70], but up to date there is no additional information related to its metabolites in these compartments. However, it is known that activated ILC2s migrate from blood into the airways of asthmatics following allergen challenge [5], and secrete endogenous $PGD_2$ [12]. Due to the known presence of ILC2s and $PGD_2$ in the airways of asthmatics, it can be hypothesized that $PGD_2$ metabolites should also be present and therefore may contribute to activation and recruiting of ILC2s via $DP_2$ signaling.

Unexpectedly, principal component analysis of all samples resulted in participant-specific rather than treatment group-specific clustering. Additional analysis revealed that these clusters are dominated by gene expression of the T Cell Receptor Delta Variable 2 and 3 (TRDV2/TRDV3), which are both used by T cells for gene rearrangement in order to generate a highly diverse T cell receptor (TCR) repertoire for broad antigen recognition [71]. Interestingly, rearranged TCR genes in ILC2s are aberrant and nonfunctional, leading to the theory that ILC2s arise from failed T cell development [72,73]. Our data emphasize the close relationship of ILC2s to T cells and suggest that there may be subsets of ILC2s based on the expression profile of TRDV genes. Further characteristic PC genes were ITGAD, RPS4Y1 and SELL, which are interestingly all involved in cell migration [74–76]. This could indicate that ILC2 subgroups exist that differ in their migration behaviour.

This study carries limitations. First, we have analyzed biological samples from only four participants. This is not uncommon when analyzing sequencing data [77], but might not be sufficient to draw generalized conclusions about $DP_2$ dependent ILC2 activity especially because principal component analysis revealed high subject-related dominance. However, with four biological replicates for each experimental group, a sequencing depth of 100 million reads per sample and the described processing of the RNA-Seq raw data, our experimental design is in line with general recommendations for RNA-Seq [78]. Second, ILC2s were isolated from blood from asthmatic patients. Because there are no data with ILC2s obtained from healthy participants, it remains unclear whether the described effects of the metabolites and $DP_2$ inhibition are specific for asthmatic derived ILC2s or characteristic for ILC2s in general. Regardless, since ILC2s were reported to be increased in the airways of asthmatics [4,5] and are known to be recruited from the blood into the human airways upon allergen challenge [5], we believe that our findings are relevant for a deeper understanding of this disease. Finally, it should be mentioned that fevipiprant is no longer being developed for the treatment of asthma. However, blockade with fevipiprant in the experiments shown here confirmed the $DP_2$ dependence of ILC2 activities and provides a deeper understanding of the $DP_2$ signaling pathway. Although fevipiprant is no longer studied in clinical trials, $DP_2$ antagonism is still a valid treatment strategy, so our findings could be of relevance to other $DP_2$ antagonists [79,80].

## Conclusions

ILC2 metabolite stimulation led to up-, while $DP_2$ inhibition led to downregulation of migration-related genes, which expression may contribute to ILC2 migration. Other DEGs and corresponding pathway analysis were related to e.g., T-cell immunobiology, pulmonary inflammation or pro-inflammatory signaling pathways and therefore could promote inflammation. Overall, our results expand our understanding of $DP_2$ initiated ILC2 activity.

## Supporting information

**S1 Fig. Gating strategy for ILC2 sorting.** After depletion of CD3-, CD14- and CD19-positive cells from PBMCs via magnetic activated cell separation (MACS), cells were stained with a PerCP-Cy5.5-labeled lineage cocktail 1 (CD4, CD8, CD14, CD16, CD19, CD34, CD123, FcεRI), a FITC-labeled lineage cocktail 2 (CD11b, CD56), CD3-BV510, CD127-BV421, CRTH2-PE and CD45-Alexa Fluor 700. **(a)** Lymphoid cells were gated by cell size (FSC-A) and cell granularity (SSC-A), and **(b)** discriminated from doublets (clotted cells) by determination of the time of flight that each cells needed to pass the filter (SSC-W). For further gating, **(d, f, g, i, j)** gates were set according to FMO controls. **(c)** CD45$^{high}$, lineage1$^-$, **(e)** lineage2$^-$, CD3$^-$, **(h)** CD127$^+$ and CRTH2$^+$ cells were defined as ILC2 and sorted into 96 U buttom well plates containing 100 μL supplemented medium. Gate boundaries were not altered between participants. (TIF)

**S2 Fig. Principal component analysis of sequenced ILC2s (unstimulated = blue, metabolites = green, fevipiprant = red).** (TIF)

**S3 Fig. Principal component analysis, depicting clustering of sequenced ILC2s (unstimulated = blue, metabolites = green, fevipiprant = red) sorted by the four ILC2 donors (a-d = subject 1–4).** (TIF)

**S1 Table. Differential gene expression analysis of genes that are related to the PPAR-γ signaling pathway using DESeq2 in PGD2 or PGD2 metabolite stimulated ILC2s.** (XLSX)

**S2 Table. Differential gene expression analysis using DESeq2 in ILC2s stimulated with PGD2 or PGD2 metabolites versus unstimulated ILC2s.** (XLSX)

**S3 Table. Differential gene expression analysis using DESeq2 in ILC2s incubated with fevipiprant versus ILC2s stimulated with PGD2 or PGD2 metabolites.** (XLSX)

**S4 Table. Gene set enrichment analysis with identified DEGs (|log2FC| ≥ 0.58, adj. p-value ≤ 0.05) of metabolite activated ILC2s versus unstimulated ILC2s using *g:Profiler*.** (XLSX)

**S5 Table. Gene set enrichment analysis with identified DEGs (|log2FC| ≥ 0.58, adj. p-value ≤ 0.05) of ILC2s incubated with fevipiprant versus ILC2s stimulated with PGD2 or PGD2 metabolites using *g:Profiler*.** (XLSX)

## Acknowledgments

The authors would like to thank the clinical staff of Fraunhofer ITEM for subject recruitment and are grateful to the study participants for their involvement in the study. Furthermore, the authors would like to thank the Helmholtz Centre for Infection Research for carrying out the sequencing.

## Author Contributions

**Conceptualization:** Meike Müller, Jens M. Hohlfeld.

**Data curation:** Christina Gress.

**Formal analysis:** Christina Gress, Maximilian Fuchs.

**Funding acquisition:** Jens M. Hohlfeld.

**Investigation:** Christina Gress, Saskia Carstensen-Aurèche.

**Methodology:** Christina Gress.

**Supervision:** Meike Müller, Jens M. Hohlfeld.

**Visualization:** Christina Gress, Maximilian Fuchs.

**Writing – original draft:** Christina Gress.

**Writing – review & editing:** Christina Gress, Saskia Carstensen-Aurèche, Meike Müller.

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
