## [Decision Letter · Decision Letter 0]

3 May 2024

PONE-D-24-10014Prostaglandin D2 receptor 2 downstream signaling and modulation of type 2 innate lymphoid cells from patients with asthmaPLOS ONE

Dear Dr. Hohlfeld,

Thank you for submitting your manuscript to PLOS ONE. After careful consideration, we feel that it has merit but does not fully meet PLOS ONE’s publication criteria as it currently stands. Therefore, we invite you to submit a revised version of the manuscript that addresses the points raised during the review process. The manuscript is focused on the "understanding of DP2 initiated ILC2 activity" (line 43). However, PGD2 and its metabolites were shown to activate other receptors, particularly PPARgamma and TP receptors, not to mention DP1. Authors dismiss the potential role of these other signaling pathways in the reported effects based on its "anti-inflammatory" (line 59) profile and the lower affinity of the metabolites (but not PGD2) compared with the affinity to DP2. Without selective DP1/TP/PPARgamma agonism/antagonism and/or expression data it is hard to dismiss offhand a potential role for this/these receptor/s in PGD2 and its metabolites on the activation of ILC2s. It is noteworthy that an essential cooperation between DP1 and DP2 in PGD2-induced LTC4 generation by eosinophils was shown before (https://doi.org/10.1111%2Fj.1476-5381.2010.01086.x). TP receptors have also been shown to activate in response to PGD2 (https://doi.org/10.1152/jappl.1989.66.4.1685, https://doi.org/10.1183/09031936.00061614) (although i don´t have any knowledge that PGD2 metabolites may have a similar effect). PPARgamma was also shown to be activated by PGD2 metabolites, although at a much higher concentration. I tend to agree that the reported effects are probably mediated by DP2, but the other possibilities should still be probably included in the introduction and most definitely in the discussion. Authors should also take into account the expression that seems to indicate that DP1 is not expressed by ILC2s while TP and PPARgamma are expressed. Authors may reconsider the assertion that differences in response to different agonists are attributable to how "ILC2 respond non-uniformly to PGD2 metabolites" and consider other receptors potentially involved and whether the impact of DP2 antagonist is due to inhibition of DP2 signaling, unopposed signaling by other receptor.

We look forward to receiving your revised manuscript.

Kind regards,

Bruno Lourenco Diaz, Ph.D.

Academic Editor

PLOS ONE

- https://doi.org/10.1038/s41598-024-51547-0

In your revision ensure you cite all your sources (including your own works), and quote or rephrase any duplicated text outside the methods section. Further consideration is dependent on these concerns being addressed.

“This research was supported by the German Center for Lung Research.”

Reviewers' comments:

Reviewer's Responses to Questions

**Comments to the Author**

1. Is the manuscript technically sound, and do the data support the conclusions?

Reviewer #1: Partly

2. Has the statistical analysis been performed appropriately and rigorously? 

Reviewer #1: Yes

3. Have the authors made all data underlying the findings in their manuscript fully available?

Reviewer #1: Yes

4. Is the manuscript presented in an intelligible fashion and written in standard English?

Reviewer #1: Yes

5. Review Comments to the Author

Reviewer #1: Thank you for the opportunity to review this manuscript. The work elegantly describes transcriptional differences in ILC2 from asthmatic patients treated with a DP2 antagonist and different PGD2 metabolites. This is relevant to the field as this lipid mediator plays an important role in the exacerbation of asthma, and has been thoroughly described as an important activator of ILC2. Overall, the paper is well written, the methodology is well executed, and the gates to identify ILC2 by FACS seem appropriate and well defined. However, some issues must be addressed by the authors.

- Major issue:

1. Delta12-PGJ2 and 15d-PGJ2 are both potent natural agonists of PPARgamma, and have been better described as such. Considering how the treatment with these specific metabolites ±fevipiprant showed strong effects that were rather different from the other treatments, it is not negligible that some of these observed effects are instead mediated by PPARgamma instead of DP2. I would strongly suggest the Authors perform one additional experiment using a PPARgamma antagonist, such as GW9662, to exclude this possibility. Given the known limitations of working with ILC2 and human samples, if this experiment is not feasible, I would suggest a directed analysis of RNAseq data, focusing on the PPARgamma signaling pathway, as a mean to assess whether this pathway is being in fact activated or not.

- Minor issues:

1. There have been some papers in the past decade describing the importance of some of these metabolites such as 15d-PGJ2 in murine models of asthma (doi: 10.3389/fimmu.2017.00740), and of PPARgamma and the activation of ILC2 (doi: 10.1038/s41385-020-00339-6; doi: 10.1038/s41467-021-22764-2). I would suggest the Authors consider mentioning these papers in the discussion, as I think it would benefit greatly from them.

2. In the discussion, lines 364 and 365, the Authors mention that “TCR expression on ILCs has not been described so far”. This is not entirely true, as there is at least one major report of such (doi: 10.1182/bloodadvances.2020002758); in fact, abortive gammadeltaTCR recombination has been taken into account as one likely theory for the origin of ILCs and their close relationship to T cells (doi: 10.3389/fimmu.2022.957711).

3. In the introduction, I would suggest the Authors to better clarify the findings of their previous publication. At times it felt like referring back to their previous work was fundamental to understanding the rationale of this paper. It is understandable that this work is a follow-up; however, minor reworks can help this body of work stand on its own, and less dependent on previous papers.

Overall, I think the paper is well written and offers interesting insight not only on the biology of ILC2 in asthma, but also on prostanoids and their importance on allergic inflammation. Other issues, such as the lack of data from healthy controls, the smaller sample size, and the fact that fevipiprant has been removed from clinical trials, have already been properly addressed by the Authors in the work itself.

6. PLOS authors have the option to publish the peer review history of their article (what does this mean?). If published, this will include your full peer review and any attached files.

Reviewer #1: **Yes: **Lukas Bolini

---

## [Author Response · Author response to Decision Letter 0]

8 Jul 2024

Dear Dr. Bruno Lourenco Diaz,

Thank you very much for your evaluation of our manuscript and the invitation to resubmit our manuscript after addressing the editor and reviewer comments below. We have carefully considered all comments and suggestions and provide you with an updated manuscript together with our point-by-point reply below. 

Furthermore, we have been asked to state the role of the funders German Center for Lung Research. Since the funders had no role, the following statement can be used as suggested by the editors: "The funders had no role in study design, data collection and analysis, decision to publish, or preparation of the manuscript."

We feel that the revision has further improved the quality of our contribution and we hope that it will meet the standard for publication in PLoS One. 

We are looking forward to your response.

Kind regards,

Jens M. Hohlfeld

Editor and Reviewer comments:

Editor

The manuscript is focused on the "understanding of DP2 initiated ILC2 activity" (line 43). However, PGD2 and its metabolites were shown to activate other receptors, particularly PPARgamma and TP receptors, not to mention DP1. Authors dismiss the potential role of these other signaling pathways in the reported effects based on its "anti-inflammatory" (line 59) profile and the lower affinity of the metabolites (but not PGD2) compared with the affinity to DP2. Without selective DP1/TP/PPARgamma agonism/antagonism and/or expression data it is hard to dismiss offhand a potential role for this/these receptor/s in PGD2 and its metabolites on the activation of ILC2s. It is noteworthy that an essential cooperation between DP1 and DP2 in PGD2-induced LTC4 generation by eosinophils was shown before (https://doi.org/10.1111%2Fj.1476-5381.2010.01086.x). TP receptors have also been shown to activate in response to PGD2 (https://doi.org/10.1152/jappl.1989.66.4.1685, https://doi.org/10.1183/09031936.00061614) (although i don´t have any knowledge that PGD2 metabolites may have a similar effect). PPARgamma was also shown to be activated by PGD2 metabolites, although at a much higher concentration. I tend to agree that the reported effects are probably mediated by DP2, but the other possibilities should still be probably included in the introduction and most definitely in the discussion. Authors should also take into account the expression that seems to indicate that DP1 is not expressed by ILC2s while TP and PPARgamma are expressed. Authors may reconsider the assertion that differences in response to different agonists are attributable to how "ILC2 respond non-uniformly to PGD2 metabolites" and consider other receptors potentially involved and whether the impact of DP2 antagonist is due to inhibition of DP2 signaling, unopposed signaling by other receptor.

Response: We thank the Editor for these valuable comments. It is right that we cannot exclude a potential role of PGD2 and its metabolites on the DP1, TP and PPAR-γ pathway without selective DP1/TP/PPAR-γ agonism/antagonism experiments. To emphasize this point, we expanded the introduction and discussion with relevant information and the proposed literature as suggested by the editor (unfortunately we have not found an appropriate reference showing TP expression on ILC2 cells). Nevertheless, as blockade of DP2 via fevipiprant showed powerful inhibitory effects, we assume that ILC2 activation via DP2 is the most prominent signaling pathway.

Changes: Introduction (line 60 & line 63-72) and discussion (line 359-377) were expanded with relevant information regarding PGD2 interaction with other receptors. The following references were added to the manuscript: 

• Mesquita-Santos et al. (Br J Pharmacol. 2011;162:1674–85. doi:10.1111/j.1476-5381.2010.01086.x) showed essential cooperation between DP1 and DP2 in PGD2-induced LTC4 generation.

• Maher et al. (Eur Respir J. 2015;45:1108–18. doi:10.1183/09031936.00061614) and Beasley et al. (J Appl Physiol (1985). 1989;66:1685–93. doi:10.1152/jappl.1989.66.4.1685) showed thromboxane (TP) receptor activation by PGD2.

• Harris & Phipps (Immunology. 2002;105:23–34. doi:10.1046/j.0019-2805.2001.01340.x) showed peroxisome proliferator activated receptor gamma (PPAR-γ) activation by PGD2 and its metabolite 15-deoxy-Δ12,14-PGJ2.

• Al Jarad et al. (Br J Clin Pharmacol. 1994;37:97–100. doi:10.1111/j.1365-2125.1994.tb04249.x) showed that activation of TP receptor results in smooth muscle contraction.

• Chen et al. (Sci Immunol. 2017;2:5196. doi:10.1126/sciimmunol.aal5196) and Xiao et al. (Mucosal Immunol. 2021;14:468–78. doi:10.1038/s41385-020-00339-6) showed that PPAR-γ signaling promotes type 2 allergic responses in mice.

• Zhang & Young (Int Immunopharmacol. 2002;2:1029–44. doi:10.1016/S1567-5769(02)00057-7), Kobayashi et al. (J Immunol. 2005;175:5744–50. doi:10.4049/jimmunol.175.9.5744) and Ercolano et al. (Nat Commun. 2021;12:2538. doi:10.1038/s41467-021-22764-2) showed that PPAR-γ signaling is involved in modulation of immune and inflammatory responses.

• McSorley & Arthus (Mucosal Immunol. 2021;14:544–6. doi:10.1038/s41385-020-00363-6) showed that PPAR-γ receptors are expressed on ILC2s and are further increased following interleukin-33 (IL-33) stimulation.

• Fujimori et al. (Gene. 2012;505:46–52. doi:10.1016/j.gene.2012.05.052) showed that PGD2 and its metabolite Δ12-PGJ2 can activate the PPAR-γ receptor.

Furthermore, additional analysis related to the PPAR-γ pathway were performed and mentioned in the manuscript (see Reviewer #1 comment [1]).

Reviewer #1

Thank you for the opportunity to review this manuscript. The work elegantly describes transcriptional differences in ILC2 from asthmatic patients treated with a DP2 antagonist and different PGD2 metabolites. This is relevant to the field as this lipid mediator plays an important role in the exacerbation of asthma, and has been thoroughly described as an important activator of ILC2. Overall, the paper is well written, the methodology is well executed, and the gates to identify ILC2 by FACS seem appropriate and well defined. However, some issues must be addressed by the authors.

Response: Thank you for this positive feedback 

Changes: none

Major issue

[1] Delta12-PGJ2 and 15d-PGJ2 are both potent natural agonists of PPARgamma, and have been better described as such. Considering how the treatment with these specific metabolites ±fevipiprant showed strong effects that were rather different from the other treatments, it is not negligible that some of these observed effects are instead mediated by PPARgamma instead of DP2. I would strongly suggest the Authors perform one additional experiment using a PPARgamma antagonist, such as GW9662, to exclude this possibility. Given the known limitations of working with ILC2 and human samples, if this experiment is not feasible, I would suggest a directed analysis of RNAseq data, focusing on the PPARgamma signaling pathway, as a mean to assess whether this pathway is being in fact activated or not.

Response: Thank you for raising this important point. As anticipated by the reviewer, it is unfortunately not feasible to perform additional ILC2 experiments because the conduct of the clinical study has been completed. However, as suggested by the reviewer, we analysed the RNAseq data with a focus on the PPAR-γ signaling pathway (KEGG: hsa03320). Accordingly,, we have screened gene expression of the receptor complex (PPAR-γ/RXR) as well as the expression of genes that are known to be induced after activation of the PPAR-γ receptor in our data set. Stimulation with PGD2 or its metabolites did not induce significant changes (defined as adj. p-value ≤ 0.05, |Log2FC| ≥ 0.58) in PPAR-γ pathway-related genes in ILC2s (S5 Table). As this is regarded an important information for the reader, we have added a supplementary table showing PPAR-γ signaling relevant genes and have added respective language to the manuscript. 

Changes: We have expanded the introduction (line 60 & line 63-72), methods (line 214-218), results (line 251-255) and discussion (line 359-377) with relevant information regarding the PPAR-γ pathway. Furthermore, we have added the analysis as supplementary table 5 (S5 Table) with the respective table description (line 766-768). As already mentioned in the response to the editor, the following references were added to the manuscript: doi:10.1111/j.1476-5381.2010.01086.x, doi:10.1183/09031936.00061614, doi:10.1152/jappl.1989.66.4.1685, doi:10.1046/j.0019-2805.2001.01340.x, doi:10.1111/j.1365-2125.1994.tb04249.x, doi:10.1126/sciimmunol.aal5196, doi:10.1038/s41385-020-00339-6, doi:10.1016/S1567-5769(02)00057-7, doi:10.4049/jimmunol.175.9.5744, doi:10.1038/s41467-021-22764-2, doi:10.1038/s41385-020-00363-6, doi:10.1016/j.gene.2012.05.052. Please refer to the reply to the editor regarding the content of the publications.

Minor issues:

[2] There have been some papers in the past decade describing the importance of some of these metabolites such as 15d-PGJ2 in murine models of asthma (doi: 10.3389/fimmu.2017.00740), and of PPARgamma and the activation of ILC2 (doi: 10.1038/s41385-020-00339-6; doi: 10.1038/s41467-021-22764-2). I would suggest the Authors consider mentioning these papers in the discussion, as I think it would benefit greatly from them. 

Response: We thank the Reviewer for this comment and appreciate the suggested literature. However, we think that the mentioned references fit already to the introduction than finally to the discussion. 

Changes: Text (line 75-77) and references (doi: 10.3389/fimmu.2017.00740; doi: 10.1016/j.clim.2004.09.008; doi: 10.1016/j.febslet.2005.11.052) were added to the introduction. 

• Coutinho et al. (Front Immunol. 2017;8:740. doi:10.3389/fimmu.2017.00740.) and Scher & Pillinger (Immunol. 2005;114:100–9. doi:10.1016/j.clim.2004.09.008.) showed anti-inflammatory properties of 15-deoxy-∆12,14-PGJ2.

• Sandig et al. (FEBS Lett. 2006;580:373–9. doi:10.1016/j.febslet.2005.11.052.) showed pro-inflammatory properties of PGD2 and its metabolites.

Changes in the manuscript related to the PPAR-γ pathway including the addition of the suggested references (doi: 10.1038/s41385-020-00339-6; doi: 10.1038/s41467-021-22764-2) are described in comment [1]. 

[3] In the discussion, lines 364 and 365, the Authors mention that "TCR expression on ILCs has not been described so far". This is not entirely true, as there is at least one major report of such (doi: 10.1182/bloodadvances.2020002758); in fact, abortive gammadeltaTCR recombination has been taken into account as one likely theory for the origin of ILCs and their close relationship to T cells (doi: 10.3389/fimmu.2022.957711). 

Response: We appreciate the comment of the reviewer and are glad to expand our discussion with this relevant information, which significantly improves our manuscript. 

Changes: Text and references (doi: 10.1182/bloodadvances.2020002758, doi: 10.3389/fimmu.2022.957711) added in line 417-422.

• Shin et al. (Blood Adv. 2020;4:5362–72. doi:10.1182/bloodadvances.2020002758.) and Kogame et al. (Front Immunol. 2022;13:957711. doi:10.3389/fimmu.2022.957711.) showed that rearranged TCR genes in ILC2s are aberrant and nonfunctional, leading to the theory that ILC2s arise from failed T cell development

[4] In the introduction, I would suggest the Authors to better clarify the findings of their previous publication. At times it felt like referring back to their previous work was fundamental to understanding the rationale of this paper. It is understandable that this work is a follow-up; however, minor reworks can help this body of work stand on its own, and less dependent on previous papers. 

Response: Thank you for this valuable comment. We are glad to expand the description of the results from our previous publication in the introduction to avoid misunderstandings. 

Changes: Added and revised text in line 79-93.

Overall, I think the paper is well written and offers interesting insight not only on the biology of ILC2 in asthma, but also on prostanoids and their importance on allergic inflammation. Other issues, such as the lack of data from healthy controls, the smaller sample size, and the fact that fevipiprant has been removed from clinical trials, have already been properly addressed by the Authors in the work itself.

Response: Thank you for this positive feedback 

Changes: none

[1] Please ensure that your manuscript meets PLOS ONE's style requirements, including those for file naming. The PLOS ONE style templates can be found at https://journals.plos.org/plosone/s/file?id=wjVg/PLOSOne_formatting_sample_main_body.pdf and https://journals.plos.org/plosone/s/file?id=ba62/PLOSOne_formatting_sample_title_authors_affiliations.pdf

Changes: Thank you for this comment. We have adapted the style of the manuscript (changes in font size and linking as heading) without using track change.

[2] We noticed you have some minor occurrence of overlapping text with the following previous publication(s), which needs to be addressed: https://doi.org/10.1038/s41598-024-51547-0

In your revision ensure you cite all your sources (including your own works), and quote or rephrase any duplicated text outside the methods section. Further consideration is dependent on these concerns being addressed.

Changes: Thank you for this comment. Unfortunately, we have not found any overlaps with our previous publication (https://doi.org/10.1038/s41598-024-51547-0) outside of the method section. We have slightly revised one similar sentence in the method section line 183-185.

[3] Thank you for stating the following financial disclosure: "This research was supported by the German Center for Lung Research."

Changes: Thank you for this comment. We have added an appropriate statement in the response to the editor (see above).

[4] Please review your reference list to ensure that it is complete and correct. If you have cited papers that have been retracted, please include the rationale for doing so in the manuscript text, or remove these references and replace them with relevant current references. Any changes to the reference list should be mentioned in the rebuttal letter that accompanies your revised manuscript. If you need to cite a retracted article, indicate the article’s retracted status in the References list and also include a citation and full reference for the retraction notice.

Changes: Thank you for this comment. Added references were mentioned in the point-by-point reply.

---

## [Decision Letter · Decision Letter 1]

11 Jul 2024

Prostaglandin D2 receptor 2 downstream signaling and modulation of type 2 innate lymphoid cells from patients with asthma

PONE-D-24-10014R1

Dear Dr. Hohlfeld,

We’re pleased to inform you that your manuscript has been judged scientifically suitable for publication and will be formally accepted for publication once it meets all outstanding technical requirements.

Kind regards,

Bruno Lourenco Diaz, Ph.D.

Academic Editor

PLOS ONE

Additional Editor Comments (optional):

Reviewers' comments:

Reviewer's Responses to Questions

**Comments to the Author**

1. If the authors have adequately addressed your comments raised in a previous round of review and you feel that this manuscript is now acceptable for publication, you may indicate that here to bypass the “Comments to the Author” section, enter your conflict of interest statement in the “Confidential to Editor” section, and submit your "Accept" recommendation.

Reviewer #1: All comments have been addressed

2. Is the manuscript technically sound, and do the data support the conclusions?

Reviewer #1: Yes

3. Has the statistical analysis been performed appropriately and rigorously? 

Reviewer #1: Yes

4. Have the authors made all data underlying the findings in their manuscript fully available?

Reviewer #1: Yes

5. Is the manuscript presented in an intelligible fashion and written in standard English?

Reviewer #1: Yes

6. Review Comments to the Author

Reviewer #1: The author addressed all the comments made in the previous round of review and I am satisfied with the implemented changes in the current version of the manuscript.

7. PLOS authors have the option to publish the peer review history of their article (what does this mean?). If published, this will include your full peer review and any attached files.

Reviewer #1: **Yes: **Lukas Bolini

---

## [Editor Report · Acceptance letter]

16 Jul 2024

PONE-D-24-10014R1 

PLOS ONE

Dear Dr. Hohlfeld, 

I'm pleased to inform you that your manuscript has been deemed suitable for publication in PLOS ONE. Congratulations! Your manuscript is now being handed over to our production team.

Kind regards, 

on behalf of

Dr. Bruno Lourenco Diaz 

Academic Editor

PLOS ONE